# Vitamin D and Visceral Obesity in Humans: What Should Clinicians Know?

**DOI:** 10.3390/nu14153075

**Published:** 2022-07-27

**Authors:** Prapimporn Chattranukulchai Shantavasinkul, Hataikarn Nimitphong

**Affiliations:** 1Division of Nutrition and Biochemical Medicine, Department of Medicine, Faculty of Medicine Ramathibodi Hospital, Mahidol University, Bangkok 10400, Thailand; prapimporn.cha@mahidol.ac.th; 2Graduate Program in Nutrition, Faculty of Medicine Ramathibodi Hospital, Mahidol University, Bangkok 10400, Thailand; 3Division of Endocrinology and Metabolism, Department of Medicine, Faculty of Medicine Ramathibodi Hospital, Mahidol University, Bangkok 10400, Thailand

**Keywords:** vitamin D, visceral obesity, adipose tissue inflammation, weight loss, caloric restriction, bariatric surgery

## Abstract

The extraskeletal effect of vitamin D on adipose tissue biology and modulation in human obesity is of great interest and has been extensively investigated. Current evidence from preclinical and clinical studies in human adipose tissue suggests that the anti-inflammatory effects of vitamin D are evident and consistent, whereas the effects of vitamin D on adipocyte differentiation, adipogenesis, and energy metabolism and the effects of vitamin D supplementation on adipokine levels are inconclusive. Interventional studies related to medical and surgical weight loss in humans have shown small or no improvement in vitamin D status. Additionally, the benefit of vitamin D supplementation for the reduction in visceral adipose tissue has only been demonstrated in a few studies. Overall, the findings on the relationship between vitamin D and visceral adipose tissue in humans are still inconclusive. Further studies are required to confirm the beneficial effects of vitamin D on ameliorating adipose tissue dysfunction.

## 1. Introduction

A well-established role of vitamin D is the regulation of calcium homeostasis and bone metabolism. Vitamin D-metabolizing enzymes and the vitamin D receptor (VDR) are widely expressed in many cells and tissues. Accordingly, numerous studies have suggested that active vitamin D or 1,25 dihydroxyvitamin D (1,25(OH)_2_D) also has many extraskeletal or non-classical effects [1,2,3]. These effects include the inhibition of cancer progression, immunomodulatory effects in certain autoimmune diseases, and effects on the cardiovascular system and obesity [1].

The prevalence of obesity has been increasing worldwide. Abdominal or visceral obesity is strongly associated with metabolic diseases, such as type 2 diabetes (T2DM), dyslipidemia, hypertension, and cardiovascular diseases [4]. Each of the different fat depots has unique characteristics, and visceral adipose tissue (VAT) is a hormonally active component of body fat that surrounds the intraabdominal organs. Visceral obesity is strongly associated with deleterious effects on cardiometabolic health because VAT has greater lipolytic potential than subcutaneous adipose tissue (SAT) [5]. Visceral obesity may result in the delivery of a greater amount of free fatty acids (FFAs) to the liver, leading to the impairment of hepatic lipid and carbohydrate metabolism [6,7]. Increased FFA flux may contribute to tissue insulin resistance and increased production of hepatic glucose and triglyceride-rich lipoproteins. This situation could contribute to the association between visceral obesity and T2DM, hepatic steatosis, hypertriglyceridemia, and an increased risk of cardiovascular diseases. Moreover, excess VAT induces low-grade chronic inflammation through the production of adipokines, such as interleukin-6 (IL-6) and tumor necrosis factor-alpha (TNF-α), and reduced adiponectin [6,7]. Excess VAT has also been proposed as a marker of SAT dysfunction as a consequence of an insufficient ability of SAT to expand through hyperplasia and play its role as a protective lipid storage organ. Therefore, excess VAT results in increased ectopic fat (lipid accumulation in the liver, heart, kidney, and skeletal muscle) [6,7,8]. Using current knowledge from preclinical and clinical studies of human adipose tissue, this review examines the role of vitamin D in regulating adipose tissue biology, focusing on adipocyte differentiation and adipogenesis, energy homeostasis, and inflammation. The current evidence relating to the effect of weight loss after dietary intervention or bariatric surgery on vitamin D status and the effect of vitamin D supplementation on the reduction in VAT is also summarized.

## 2. Methodology and Literature Search

This is a narrative review. To summarize the current evidence for vitamin D and visceral obesity in humans, PubMed and Google Scholar were searched to find relevant articles published between 1971–2022. The following combinations of keywords were used: vitamin D OR free vitamin D OR vitamin D action OR vitamin D metabolism OR classical action of vitamin D OR non-classical action of vitamin D OR genomic action of vitamin D OR non-genomic action of vitamin D OR vitamin D receptor OR vitamin D deficiency OR vitamin D insufficiency OR obesity OR overweight OR body mass index OR fat mass OR abdominal obesity OR visceral adipose tissue OR subcutaneous adipose tissue OR human adipose tissue OR human preadipocyte OR human adipocyte OR adipocyte differentiation OR adipogenesis OR adipose tissue function OR lipolysis OR lipogenesis OR lipid metabolism OR energy metabolism OR inflammation OR inflammatory cytokines OR adipokines OR cytokine OR immune response OR leptin OR adiponectin OR TNF-α OR C-reactive protein OR interleukin OR insulin secretion OR insulin resistance OR glucose homeostasis OR HOMA-B OR HOMA-IR OR weight loss OR bariatric surgery OR gastric bypass surgery OR vitamin D supplementation AND adipose tissue OR vitamin D supplementation AND weight loss OR vitamin D supplementation AND inflammation. We mainly included preclinical and clinical studies in humans. The included studies were randomized controlled trials (RCT), observational studies, systematic reviews, meta-analyses, and review articles that were published as full text. For vitamin D supplementation studies in humans, we included trials that used native vitamin D. The exclusion criteria were as the follows: case report studies, only abstract available and used active vitamin D in studies of vitamin D supplementation in human. 

## 3. Vitamin D Metabolism

Vitamin D is a multifunctional hormone or prohormone because of its contribution to many processes in many cells and tissues. The metabolism and classical and non-classical functions of vitamin D are shown in Figure 1. For most people, the main source of vitamin D is skin exposure to sunlight [2]. After exposure to ultraviolet-B (290–315 nm), ultraviolet-B photons cause the photolysis of 7-dehydrocholesterol (7-DHC, provitamin D_3_) to previtamin D_3_, which is thermally isomerized (37 °C) to vitamin D_3_ [3]. Vitamin D can also be obtained from the diet and supplementation (vitamin D_2_ and D_3_ forms). In the liver, vitamin D (“D” represents D_2_ and/or D_3_) is hydroxylated to 25-hydroxyvitamin D [25(OH)D] by 25-hydroxylases (*CYP2R1*, *CYP27A1*, *CYP3A4*, and *CYP2J2*). Then, in the kidney, 25(OH)D is converted to 1,25(OH)_2_D by 1α-hydroxylase (*CYP27B1*). 24-Hydroxylase (*CYP24A1*) plays a major role in the degradation of both the active form, 1,25(OH)_2_D, and 25(OH)D. Vitamin D also has autocrine/paracrine functions in many tissues. For example, local activation of vitamin D can occur in adipose tissue because of the expression of 1α-hydroxylase in this tissue [1].

The total serum 25(OH)D level is considered the best indicator of vitamin D status. The optimum serum 25(OH)D level for musculoskeletal health, especially in people with osteoporosis, is still controversial, and it ranges between 20 and 30 ng/mL [9,10,11,12,13]. Most biological actions of the active form of vitamin D are exerted through the nuclear vitamin D receptor, VDR. A heterocomplex of ligand-bound VDR and retinoid X acid receptor translocates into the nucleus, binds to the vitamin D response elements of genes, and leads to gene transcription. The VDR is also localized in the plasma membrane and exerts rapid membrane-initiating responses, which are considered non-genomic actions of vitamin D. The VDR is also present in mitochondria. 1,25(OH)_2_D acts through the mitochondrial VDR to regulate cellular differentiation and biosynthesis pathways, including lipid biosynthesis [14,15].

Vitamin D binding protein (VDBP) is the specific chaperone for vitamin D and its metabolites. VDBP facilitates the delivery of vitamin D to various storage sites and target cells and tissues. Therefore, VDBP influences the total amount of vitamin D available to those cells and tissues. Approximately 85–90% of 25(OH)D and 1,25(OH)_2_D are bound to VDBP. Less than 15% of these vitamin D metabolites are weakly bound to albumin, and less than 1% of circulating vitamin D is in the free form [16]. The association between 25(OH)D and VDBP is crucial for the renal handling of 25(OH)D and endocrine synthesis of 1,25(OH)_2_D. However, unbound, free 25(OH)D drives many of the non-classical actions of vitamin D, such as the regulation of hormone and cytokine secretion, immune function, and cellular proliferation and differentiation [17].

## 4. Association between Vitamin D and Adipose Tissue

Adipose tissue is the major storage site of vitamin D (D_2_ and D_3_) and 25(OH)D [18,19,20]. A total of 65% of vitamin D is present as native vitamin D, and the rest is 25(OH)D [19]. However, the relationship between adipose storage of vitamin D and serum 25(OH)D levels in people with obesity is complex and has not been completely determined. A study in women reported that, after Roux-en-Y gastric bypass, VAT had approximately 20% more vitamin D per gram than SAT [21]. Serum 25(OH)D concentrations were strongly associated with adipose tissue concentrations [21,22], while vitamin D_3_ concentrations in adipose tissue varied and were not associated with serum 25(OH)D levels [23].

Importantly, enzymes involved in vitamin D metabolism, including 25-hydroxylase (*CYP2R1*, *CYP27A1*, and *CYP2J2*), 1α-hydroxylase, and 24-hydroxylase, are expressed in SAT and VAT [24,25,26] (Figure 1). Several genes involved in adipocyte differentiation (e.g., peroxisome proliferator-activated receptor γ (*PPAR**-**γ*), adipocyte-binding protein 2 (*AP2*) or fatty acid-binding protein (FABP4), lipoprotein lipase (*LPL*), and thioredoxin (*Trx*)) and function (e.g., *IL**-1*, *IL**-6*, monocyte chemoattractant protein 1 (*MCP**-1*), and NADPH oxidase (*NOX*)) are regulated by vitamin D [27,28,29,30]. Notably, vitamin D-metabolizing enzymes and VDR expression were different between lean subjects and those with obesity and between SAT and VAT. A study showed that mRNA expression levels of *CYP2J2* and *CYP27B1* were reduced in the SAT of subjects with obesity compared with those of lean subjects [25]. Additionally, VDR gene expression was higher in subjects with morbid obesity than in those with a lower body mass index (BMI) [26]. VAT expresses higher CYP27A1 mRNA levels and lower CYP27B1 and CPY2J2 mRNA levels than SAT [25]. Additionally, 1,25(OH)_2_D_3_ increases VDR gene expression in adipose tissue from subjects with obesity but not from lean subjects [26]. These data suggest variations in local metabolism and effects of vitamin D in adipose tissue, depending on the grading of obesity and type of adipose tissue. Adipose tissue plays an important role in vitamin D metabolism, leading to vitamin D inadequacy in individuals with obesity. Additionally, an association may exist between vitamin D deficiency and adipose tissue-related metabolic disorders.

## 5. Vitamin D Action in Human Adipose Tissue

A growing body of evidence has suggested that vitamin D is involved in numerous processes in human adipose tissue. This section summarizes the evidence, mainly from studies of human adipocytes and adipose tissue, related to the effects of vitamin D on adipogenesis, energy homeostasis, and inflammation. These effects of vitamin D are mediated through the VDR. We refer to comprehensive reviews to discuss the molecular mechanisms of vitamin D in adipose tissue differentiation and function in different cell types and species [14,31,32].

### 5.1. Role of Vitamin D in Adipocyte Differentiation and Adipogenesis

Inconsistent results on the specific roles of vitamin D on adipocyte differentiation and adipogenesis in different cell types have been reported. Most studies of 3T3-L1 cells (mouse preadipocytes) reported that 1,25(OH)_2_D_3_ inhibited adipocyte differentiation [33,34]. However, other studies showed that 1,25(OH)_2_D_3_ promoted adipocyte differentiation in bone marrow-derived mesenchymal stem cells of pigs [27] and mice [30]. To date, a few studies of primary human adipose-derived stem cells (hASCs) and primary human subcutaneous preadipocytes have shown that vitamin D enhances adipocyte differentiation and lipid accumulation [24,30] (Table 1). In addition, a study of primary human subcutaneous preadipocytes showed that 25(OH)D_3_ promoted the differentiation of human adipocytes (Table 1), most likely via its activation with 1,25(OH)_2_D_3_ [24]. To date, no studies have examined the effect of vitamin D on the differentiation of human visceral adipocytes. These inconclusive outcomes might be due to differences in the cell type, the stage of differentiation, and the period of vitamin D addition [14,31,32]. Vitamin D may promote adipocyte differentiation in cells that are at an advanced stage of differentiation, such as human preadipocytes.

### 5.2. Role of Vitamin D in Energy Homeostasis

Vitamin D is involved in energy homeostasis, but the exact mechanism has not yet been determined. Evidence from VDR and CYP27B1 knockout mice suggests that 1,25(OH)_2_D_3_ plays an important role in lipid metabolism. These mice had a lean phenotype with a reduction in body fat and serum leptin levels and an increase in total energy expenditure (TEE), oxygen consumption, and CO_2_ production [15,35,36]. A recent study in rats demonstrated that vitamin D-deficient rats had a significant decrease in food intake, homeostasis model of assessment for insulin resistance (HOMA-IR), irisin levels, and the respiratory quotient (RQ), which suggest increased lipid use [37]. In this study, negative associations between irisin levels and HOMA-IR and the RQ were observed in vitamin D-deficient rats [37]. Therefore, these novel findings suggest a relationship between vitamin D and irisin, which possibly affects TEE and body weight. Unfortunately, patients with rickets, which is a disease caused by 1α-hydroxylase deficiency and VDR resistance, have not been investigated for any alterations in lipid metabolism [15]. Accordingly, evidence on vitamin D and lipid metabolism in humans is still lacking.

Emerging evidence from studies in cells and animal models suggests that vitamin D improves glucose homeostasis and promotes insulin sensitivity by promoting beta-cell function and increasing peripheral insulin sensitivity [38]. The established mechanisms of this improvement are ameliorating deleterious molecular mechanisms involved in the pathophysiology of beta-cell dysfunction, lowering oxidative damage, suppressing inflammatory responses, and promoting insulin receptor substrate (IST) expression and activity [38]. Additionally, some preclinical studies reported that vitamin D modulated leptin and adiponectin, which are adipokines produced in adipose tissue, and they play a role in glucose and lipid metabolism [31]. To determine the role of vitamin D in energy homeostasis in humans, clinical studies are required. Unfortunately, randomized controlled trials (RCTs) in humans are limited (Table 2). Studies with a small number (<100) of participants with overweight or obesity and a high risk of diabetes or T2DM showed that vitamin D_3_ supplementation improved insulin secretion indices, including the disposition index [39] and homeostasis model of assessment of β-cell activity (HOMA-%B) [40]. A reduction in oxidative stress was shown in a study of overweight pregnant women with gestational diabetes (GDM). In this study, vitamin D_3_ and calcium supplementation significantly increased plasma total glutathione (GSH; an antioxidant) concentrations and prevented a rise in plasma malondialdehyde (MDA; a biomarker of oxidative stress) concentrations compared with placebo [41]. However, studies in humans have reported inconsistent results on the association between vitamin D and adiponectin. In one study in differentiated human adipocytes, 1,25(OH)_2_D_3_ did not alter adiponectin expression [42]. In contrast, in another study, vitamin D_3_ supplementation in overweight/obese and vitamin D-deficient adults resulted in higher adiponectin and leptin levels than in those with no treatment [43]. Therefore, more clinical studies are required to confirm the beneficial role of vitamin D in energy homeostasis in people with obesity.

### 5.3. Role of Vitamin D in Inflammation

It has been discovered that all cells involved in the innate and adaptive immune systems express the VDR and CYP27B1 and modulate the local synthesis of calcitriol, which may have an immunomodulatory effect on different cytokines [1]. Most in vitro studies have shown that 1,25(OH)_2_D_3_ decreases gene expression, proinflammatory cytokine levels (e.g., IL-1β, IL-1α IL-6, and TNF-α), and Toll-like receptor 2 (TLR2) and TLR4 protein and mRNA levels in human monocytes/macrophages [44,45,46,47]. Similarly, most studies in preadipocytes (mostly isolated from SAT) and adipocytes showed that 1,25(OH)_2_D_3_ reduced MCP-1, IL-6, IL-8, RANTES, IL6, and IL-1β expression [48,49,50]. Few studies have examined the anti-inflammatory effects of 1,25(OH)_2_D_3_ in adipose tissue. A recent study in humans [51] investigated the effects of 1,25(OH)_2_D_3_ on adipokine levels in visceral and subcutaneous adipose tissues. This study reported that treatment with 1,25(OH)_2_D_3_ for 2 days decreased leptin and IL-6 in both depots. Interestingly, in primary culture of human adipocytes, treatment with 25(OH)D_3_, but not vitamin D_3_, suppressed adipokine expression. These findings suggest that *CYP27B1* is expressed and can convert 25(OH)D_3_ to 1,25(OH)_2_D_3_ in human adipocytes [51]. Another study assessed oxidative stress in VAT and vascular samples after in vitro administration of 1,25(OH)_2_D_3_. The main findings were that 1,25(OH)_2_D_3_ treatment in patients with obesity decreased oxidative stress and improved vascular function [52].

Studies that focused on the association between 25(OH)D levels and proinflammatory cytokine levels in humans show inconsistent results (Table 3). A cross-sectional study demonstrated that lower 25(OH)D levels were associated with higher plasma IL-6 and TNF-α levels in participants with normal weight and higher plasma adiponectin levels in participants who were overweight [53]. A recent study of a 1-year lifestyle intervention program with no vitamin D supplementation reported a significant reduction in VAT and leptin levels and an increase in 25(OH)D concentrations [54]. In this study, a 1-year increase in 25(OH)D levels were independently correlated with a 1-year decrease in leptin levels after adjustment for the change in adiposity [54]. These findings suggest an association between leptin levels and vitamin D status. Similarly, in vivo studies of the benefit of vitamin D treatment and the modulation of inflammatory responses in humans are inconclusive (Table 3). In one in vivo study, vitamin D supplementation in participants with obesity and vitamin D deficiency did affect changes in circulating and mRNA levels of inflammatory markers in subcutaneous abdominal adipose tissue compared with no vitamin D treatment [25]. However, another study of 1-year supplementation of vitamin D_3_ reported a non-significant reduction in serum IL-6 levels, but there was a significant increase in C-reactive protein (CRP) levels. Additionally, there were no changes in insulin resistance or TNF-α concentrations [55]. The results from systematic reviews and meta-analyses are also inconclusive. A systematic review with a meta-analysis of subjects with overweight or obesity reported that supplementation with vitamin D did not decrease CRP, TNF-α, or IL-6 levels [56]. However, a meta-analysis of RCTs in patients with T2DM and overweight or obesity reported that vitamin D supplementation significantly decreased circulating CRP concentrations but did not affect TNF-α or IL-6 concentrations [57]. Notably, the variability in vitamin D dosage and duration and the differences in the characteristics of participants might have led to inconclusive results.

## 6. Vitamin D and Obesity

Vitamin D insufficiency and deficiency are highly prevalent in patients with obesity, and several mechanisms could be involved [58,59]. These mechanisms include inadequate sunlight exposure, volumetric dilution, decreased vitamin D synthesis in the skin, and decreased bioavailability of fat-soluble vitamin D due to sequestration into adipose tissue.

Much evidence from observational studies has shown an inverse association between vitamin D status and obesity [60,61,62,63], particularly in patients with visceral obesity [64,65,66,67,68,69]. However, the causal relationship between vitamin D deficiency and visceral obesity needs to be determined. Moreover, the effect of weight loss after dietary intervention or bariatric surgery on improving vitamin D status is unclear. Additionally, the effect of vitamin D supplementation on weight loss is inconclusive.

### 6.1. Effect of Medical and Surgical Weight Loss on Vitamin D Status

One of the mechanisms of vitamin D deficiency and insufficiency in obesity is the sequestration of vitamin D into adipose tissue [58]. Therefore, reduced body fat mass after extensive weight loss results in increased vitamin D release from adipose tissue and increased vitamin D bioavailability from decreased adipose sequestration. However, the effect of weight loss from lifestyle and dietary interventions on vitamin D status is small, while the effect of weight loss surgery on 25(OH)D levels is inconsistent.

Treatment of obesity always begins with comprehensive lifestyle modification, which is a combination of caloric restriction, exercise, and behavior therapy. However, the effect of weight loss from lifestyle and dietary interventions on vitamin D status is still inconclusive. A recent meta-analysis indicated that weight loss after caloric restriction and/or exercise interventions was associated with a small increase in serum 25(OH)D levels [70]. However, another study did not demonstrate a significant improvement in vitamin D status after dietary intervention-induced weight and fat mass reduction [71] (Table 4).

Bariatric or metabolic surgery is recognized as the most effective treatment for severe obesity. This surgery results in a dramatic decline in body weight and sustained weight loss, as well as the resolution of obesity-related complications, including T2DM. Nevertheless, vitamin and mineral deficiencies are well-known complications after bariatric surgery. Moreover, preoperative micronutrient deficiencies, such as vitamin D, iron, and folate deficiencies, are common in individuals with obesity. The mechanisms of vitamin D deficiency after bariatric surgery can be explained by reduced food intake, low sun exposure, and fat malabsorption from a malabsorptive procedure. However, the effect of bariatric surgery on vitamin D status is challenging to assess owing to the variability in the type and technique of bariatric surgery and the dose and form of vitamin D supplementation after bariatric surgery. Moreover, patients’ adherence to follow-up visits and vitamin supplementation varies. Previous studies have shown conflicting results regarding the effect of bariatric surgery on vitamin D status, with either increased [72,73,74], decreased [75,76], or unchanged [23,77,78,79,80] 25(OH)D levels. We summarized the data from systematic reviews and meta-analyses on the effect of medical and surgical weight loss on vitamin D status [70,71,74,79,80,81,82] (Table 4).

Overall, the findings on vitamin D status after medical and surgical weight loss are inconclusive because of the high heterogeneity and the small number of studies, as well as various doses and forms of vitamin D supplementation after bariatric surgery. Further studies are required to confirm the effect of weight loss on vitamin D status.

### 6.2. Effect of Vitamin D Supplementation on Weight Reduction and Visceral Fat Loss

An animal study showed that vitamin D supplementation could lead to weight loss, reduced visceral fat, and lowered leptin and plasma TNF-α concentrations [83]. However, only a few clinical studies have examined the effect of vitamin D supplementation on visceral fat reduction, and they showed inconsistent findings. Some clinical trials demonstrated a benefit of vitamin D supplementation in reducing total body fat and/or VAT [84,85,86]. However, several studies did not show a positive effect of vitamin D supplementation on visceral fat reduction [87,88,89]. A meta-analysis [90] of 11 RCTs, including 947 participants with obesity, showed that vitamin D_3_ supplementation (doses varied from 25,000 to 600,000 IU/month, follow-up from 1 to 12 months) did not affect weight loss. However, the BMI and waist circumference (WC) were significantly reduced with vitamin D_3_ supplementation. Another meta-analysis [91] of 20 RCTs and 1146 participants investigated the effect of vitamin D-fortified food for 2–24 months. This meta-analysis showed a significant reduction in the WC and the waist-to-hip ratio (WHR) but no effect on weight, BMI, fat mass, or lean mass. A recent meta-analysis [89] of 20 RCTs involving 3153 healthy participants assessed the effect of vitamin D supplementation on the BMI, WC, and WHR. This meta-analysis showed that vitamin D supplements (doses varied from 100 to 8571 IU/day and the duration ranged from 1.5 to 36 months) had no significant effect on the BMI, WC, and WHR compared with placebo. However, a significant reduction in the BMI and WC was only found in the subgroup of women in studies conducted in Asia and when the duration of vitamin D supplementation was longer than 6 months. These discrepancies among studies could be due to the various study designs (some studies combined vitamin D supplementation with caloric restriction and/or exercise), populations with different comorbid diseases, the dose, form, and duration of vitamin D supplementation, season, and the region of the enrolled subjects, which can directly affect cutaneous vitamin D synthesis. Moreover, genetic polymorphisms [92] and the patient’s baseline BMI [93] appear to affect the vitamin D response. Further studies are required to assess the beneficial effect of vitamin D supplementation on visceral fat loss.

## 7. Discussion

The relationship between vitamin D and visceral fat in humans is complex and not completely elucidated. This review summarizes relevant preclinical and clinical studies in humans. We provide insights into issues ranging from the molecular aspects of vitamin D and adipose tissue biology to clinical aspects of vitamin D and visceral adiposity and weight loss. Many studies suggested that vitamin D plays an important role in regulation of inflammatory response in adipose tissue. However, vitamin D supplementation is not consistently decrease inflammatory markers. Growing evidence also suggested the effects of vitamin D on adipocyte differentiation and adipogenesis whereas the effect of vitamin D on energy metabolism is still lacking. To date, only a few and mostly underpowered randomized clinical trials have been conducted to test the effectiveness of vitamin D supplementation in facilitating weight loss. 

From our perspective, there are considerations in the interpretation of studies related to a relationship between vitamin D and visceral obesity in human. The differences in the grading of obesity and types of adipose tissue possibly explained the variations in local metabolism of vitamin D in adipose tissue and effects of vitamin D in modulating adipose tissue activity. Since modalities of obesity treatment influence on the magnitude of weight loss and vitamin D malabsorption. These issues could partly lead to an inconsistent result related to an improvement in vitamin D status after weight reduction. Few aspects about the effects vitamin D on VAT, for example, the effect of vitamin D on the differentiation of visceral adipocytes, are lacking.

This review has several limitations. The included studies were heterogeneous in terms of the study design, the characteristics of the participants, and the vitamin D dosage and duration. In addition, this is not a systematic review.

## 8. Conclusions

The current review summarizes recent findings on the role of vitamin D in human adipose tissue biology, focusing on adipocyte differentiation and adipogenesis, energy homeostasis, inflammation, and the clinical significance of vitamin D and human visceral obesity. Preclinical studies support an important role of vitamin D in adipose tissue biology. However, the effects of vitamin D on adipocyte differentiation, adipogenesis, and energy metabolism in humans have not been fully determined, and inconsistent results have been reported. Nevertheless, the available evidence regarding the association between vitamin D and anti-inflammatory effects is more consistent. Some studies, but not all, have shown a reduction in proinflammatory mediators after receiving vitamin D supplementation. Although epidemiological studies have shown an inverse association between vitamin D adequacy and the prevalence of obesity, the results from intervention studies of the effect of vitamin D supplementation on visceral fat loss are still inconclusive. Further studies are required to determine the benefit of vitamin D supplementation on the improvement of adipose tissue dysfunction.

## Figures and Tables

**Figure 1 nutrients-14-03075-f001:**
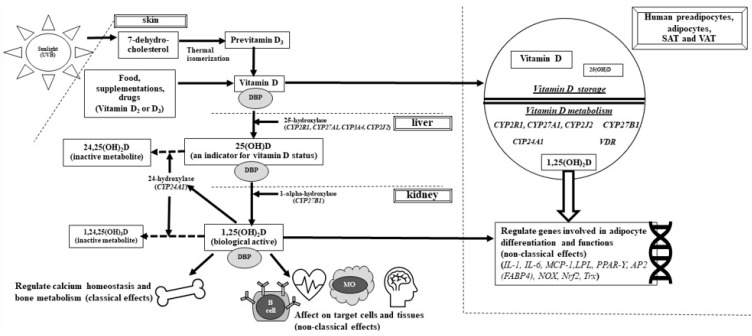
**Vitamin D metabolism and functions**. Vitamin D (synthesis from skin and intake from food, supplementation, and drugs) are activated by 2 hydroxylation processes; 25-hydroxylation in the liver and 1-alpha-hydroxylation in the kidneys, and then becomes an active form [1,25(OH)_2_D]. 24-hydroxylase inactivates both 1,25(OH)_2_D and 25(OH)D to inactive metabolites. DBP facilitates delivering vitamin D and its metabolites to various target cells and tissues. Many cells and tissues, including human preadipocytes, adipocytes, SAT and VAT express vitamin D metabolizing enzymes and can activate 25(OH) D to an active form. 1,25(OH)_2_D also regulates many genes involved in adipocyte differentiation and functions. D: D_2_ or D_3_, DBP: vitamin D binding protein, 25(OH)D: 25-hydroxyvitamin D, 1,25(OH)_2_D: 1,25 dihydroxyvitamin D, *VDR*: vitamin D receptor *IL:* interleukin, *MCP**-**1*: monocyte chemoattractant protein 1, *LPL*: lipoprotein lipase, *PPAR**-**γ*: peroxisome proliferator-activated receptor γ, *AP2*: adipocyte-binding protein 2, *FABP4*: fatty acid-binding protein 4, *NOX:* NADPH oxidase, *Nrf2*: transcription nuclear factor 2, *Trx:* thioredoxin, SAT: subcutaneous adipose tissue, VAT: visceral adipose tissue.

**Table 1 nutrients-14-03075-t001:** Effects of 25(OH)D and 1,25(OH)_2_D treatments on human adipocyte differentiation and adipogenesis.

Cell Type	Type of Vitamin D, Dose, Duration	Results
hASCs derived from women with normal BMI [30]	1,25(OH)_2_D_3_, 10 nM, 7 and 14 days	- Increased the expression of FABP4, FASN, and PPAR-γ mRNA - Increased ACC, FABP4, and FASN protein levels - Promoted lipid accumulation
Human subcutaneous preadipocytes derived from obese men and women [24]	1,25(OH)_2_D_3_, 0.1 and 10 nM, 14 days	- Increased the expression of PPAR-γ and LPL mRNA - Increased FABP protein levels - Increased triglyceride accumulation
Human subcutaneous preadipocytes derived from obese men and women [24]	25(OH)D_3_, 1 and 10 nM, 14 days	- Increased the expression of LPL mRNA - Increased FABP4 protein levels - Increased triglyceride accumulation

1,25 (OH)_2_D_3_: 1,25-dihydroxyvitamin D3; 25(OH)D_3_: 25-hydroxyvitamin D_3_; ACC: acetyl-CoA carboxylase; BMI: body mass index; FABP4: fatty acid-binding protein 4; FASN: fatty acid synthase; hASCs: human adipose-derived stem cells; LPL: lipoprotein lipase; PPAR-γ peroxisome proliferator-activated receptor γ.

**Table 2 nutrients-14-03075-t002:** Effects of vitamin D supplementation on β-cell function, insulin sensitivity, and adipokines related to energy homeostasis in humans.

Participants	Type of Vitamin D, Dose, Duration	Results
96 participants with prediabetes or with newly diagnosed type 2 diabetes [39]	Vitamin D_3_, 5000 IU/day vs. placebo, 6 months	- Increased M-value (a marker of peripheral insulin sensitivity) in the vitamin D group vs. stable in the placebo group (*p* = 0.009) - Improved β-cell function (disposition index) in the vitamin D group vs. stable in the placebo group (*p* = 0.039)
20 participants with type 2 diabetes [40]	Vitamin D_3_, 5000 IU/day vs. placebo, 12 weeks	- Increased HOMA-%B in the vitamin D group (*p* = 0.03) * vs. non-significant increased HOMA-%B in the placebo group (*p* = 0.08) *
56 women with gestational diabetes (at 24–28 weeks of gestation) [41]	Vitamin D_3_ 50,000 IU at baseline and day 21 + calcium 1000 mg/day vs. placebo, 3 weeks	- Decreased HOMA-IR in the treatment group vs. stable in the placebo group (*p* = 0.001) - Increased QUICKI in the treatment group vs. stable in the placebo group (*p* = 0.003) - A significant increase in GSH in the treatment group when compared with the placebo group (*p* = 0.03) - A smaller increase in MDA in the treatment when compared with the placebo group (*p* = 0.03)
54 participants with obesity and vitamin D deficiency (25(OH)D < 20 ng/mL) [43]	Vitamin D_3_ 100,000 IU bolus, then 4000 IU/day vs. placebo, 16 weeks	- Greater increases in adiponectin (*p* = 0.002) and leptin (*p* = 0.002) in the vitamin D group when compared with the placebo group ^a^

25(OH)D: 25-hydroxyvitamin D; GSH: glutathione; HOMA-% B: homeostasis model of assessment of β-cell activity; HOMA-IR: homeostasis model of assessment for insulin resistance; MDA: malondialdehyde: QUICKI: quantitative insulin sensitivity check index; *, different within group; ^a^, after adjustment for baseline 25(OH)D levels, season, sun exposure, and dietary vitamin D intake.

**Table 3 nutrients-14-03075-t003:** The association between 25(OH)D levels and inflammatory cytokines and the effect of vitamin D supplementation on inflammatory cytokines in humans.

**The Association between 25** **(** **OH** **)** **D Levels and Inflammatory Cytokines**		
**Study Design, n**	**Results**	
A cross-sectional population-based study, 281 [53]	- A negative association between plasma IL-6 and TNF-α levels and serum 25(OH)D concentration in normal-weight participants - A negative association between plasma adiponectin level and serum 25(OH)D concentration in overweight participants	
Post hoc analysis from 1-year lifestyle intervention program, 113 men [54]	- An increase in 25(OH)D levels were associated with a decrease in leptin levels after adjustment for changes in adiposity - No association between changes in 25(OH)D levels and changes in adiponectin levels	
**The Effect of Vitamin D Supplementation on Inflammatory Cytokines**		
**Study design, n**	**Type of vitamin D, dose, duration**	**Results**
RCT, subcutaneous abdominal adipose tissue from 40 participants with obesity and vitamin D deficiency (25(OH)D < 20 ng/mL) [25]	Vitamin D_3_ 7000 IU/day vs. placebo, 26 weeks	- No differences in the changes in MCP-1, IL-6, IL-8, and adiponectin levels from baseline between the 2 groups - No differences in the expression levels of MCP-1, IL-6, and IL-8 before and after treatment with either placebo or vitamin D
RCT, 332 participants with overweight and obesity [55]	Vitamin D_3_ 40,000 IU/week vs. 20,000 IU/week vs. placebo, 1 year	- A non-significant decrease in IL-6 (*p* = 0.08) and a significant increase in CRP (*p* < 0.05) in the vitamin D group when compared with the placebo group - No effect of vitamin D supplementation on TNF-α levels
A systematic review and meta-analysis of 13 RCTs, 1955 participants with obesity or overweight [56]	Vitamin D_3_ 700–200,000 IU/day or vitamin D_2_ 150,000 IU at 0 and 12 weeks, duration 4–156 weeks (mean 41 weeks)	- No significant reduction in CRP, TNF- α, and IL-6 levels after receiving vitamin D supplementation
A systematic review and meta-analysis of 13 RCTs, 875 participants with type 2 diabetes [57]	Vitamin D_2_ or D_3_ 20–6000 IU/day or 25,000 or 50,000 IU/week, duration 8–52 weeks (median 12 weeks)	- A significant decrease in CRP in the vitamin D group when compared with no vitamin D treatment (*p* = 0.005) - No effects of vitamin D supplementation on TNF-α and IL-6

25(OH)D: 25-hydroxyvitamin D; CRP: C-reactive protein; IL-6: interleukin 6; IL-8: interleukin 8; MCP-1: monocyte chemoattractant protein-1; RCT: randomized-controlled trial; TNF-α: tumor necrosis factor-alpha.

**Table 4 nutrients-14-03075-t004:** The effect of medical and surgical weight loss on vitamin D status.

Study Design, n	Weight Loss Intervention	Results
A systematic review and meta-analysis of 15 clinical trials (4 RCTs and 11 non-RCTs), 3471 participants with obesity and overweight [70]	Caloric restriction and/or exercise intervention without weight loss medications ± vitamin D supplementation (median vitamin D intake was 350 IU/day) vs. weight maintenance, first follow-up visit at 6–104 weeks (median 26 weeks)	Weight loss was associated with a small but significant increase in 25(OH)D levels (mean difference 3.76 nmol/L, 95% CI: 2.38, 5.13 nmol/L).
A systematic review and meta-analysis of 23 clinical trials (14 RCTs and 9 single-arm studies), 2085 participants with obesity and overweight [71]	Caloric restriction and/or exercise intervention without vitamin D supplementation vs. weight maintenance, study duration 2 weeks to 2 years	Weight loss was not significantly associated with increased 25(OH)D levels (6.0 nmol/L, 95% CI: −12.42, 0.47 in the weighted mean difference of 25(OH)D for weight loss of 10 kg (*p* = 0.06)).
A systematic review and meta-analysis of 7 studies (2 RCTs and 3 observational studies), 4282 cases/15,630 controls participants with obesity [80]	Bariatric surgery (RYGB or DS with or without BPD) compared to non-surgical controls, 1 year postoperative	25(OH)D levels did not change significantly compared to controls (weight mean difference 6.79%, 95% CI: −9.01, 22.59).
A systematic review and meta-analysis of 10 prospective studies, 344 participants with morbid obesity [79]	RYGB, vitamin D, and calcium supplementation after surgery, 6–36 months postoperative	25(OH)D levels did not increase significantly after RYGB compared to baseline levels despite vitamin D supplementation (mean difference 1.35 ng/mL, 95% CI: −1.12, 3.83).
A systematic review and meta-analysis of 12 studies (6 RCTs and 6 single-arm studies), 1285 participants with obesity [74]	Bariatric surgery with vitamin D supplementation compared with different types of bariatric surgery or lifestyle intervention, 1 year postoperative	25(OH)D levels increased significantly after surgery, and the prevalence of vitamin D deficiency decreased only in RCTs with vitamin D supplementation >800 IU/day (prevalence of vitamin D deficiency was 54% before surgery and 31% after surgery).
A systematic review and meta-analysis of 13 studies (2 RCTs, 9 observational studies), 1503 participants with morbid obesity [81]	Bariatric surgery (RYGB or SG), 1–5 years postoperative	25(OH)D levels were significantly lowered in patients who underwent RYGB compared to SG at 1 year postoperative (mean difference −1.85 ng/mL, 95% CI: −3.32, −0.39).
A systematic review and meta-analysis of 5 studies in participants with morbid obesity receiving sufficient vitamin D supplementation according to guidelines [82]	Bariatric surgery (RYGB or SG), 3 months–5 years postoperative	Vitamin D levels significantly increased after RYGB (weighted mean difference 22.71 ng/mL; 95% CI, 15.87, 29.56 at 6–11 month) and SG (weight mean difference 6.03 ng/mL; 95% CI, 4.18, 7.89 at 12–23 months).

25(OH)D: 25-hydroxyvitamin D; RCT: randomized-controlled trial; CI: confidence interval; RYGB: Roux-en-Y gastric bypass; DS: duodenal switch; BPD: biliopancreatic diversion; SG: sleeve gastrectomy.

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
