# Peer review of "Vitamin D and Visceral Obesity in Humans: What Should Clinicians Know?"

_nutrients, 2022, doi:10.3390/nu14153075_

Round 1

Reviewer 1 Report

Dear authors this is a very interesting and well-written narrative review. I would suggest that you add a section Methodology, in which you will explain the methodology and the strategy used for the articles assessment. Furthermore, you could add a discussion section in which you will critically discuss and synthesise all your findings.

Author Response

July 18, 2022

Dear reviewer,

We greatly appreciate the opportunity to revise and resubmit our manuscript “Vitamin D and Visceral Obesity in Human: What Should Clinicians Know?”

We have carefully considered useful suggestions from the Editor and Reviewers, and revised the manuscript accordingly. We have provided detailed responses to the Reviewer’s comments below.

Journal: Nutrients (ISSN 2072-6643)

Manuscript ID: nutrients-1811554

Type: Review

Title: Vitamin D and Visceral Obesity in Humans: What Should Clinicians Know?

Authors: Prapimporn Chattranukulchai Shantavasinkul , Hataikarn Nimitphong *

Reviewer 1 comment

Comments and Suggestions for Authors

Dear authors this is a very interesting and well-written narrative review. I would suggest that you add a section Methodology, in which you will explain the methodology and the strategy used for the articles assessment. Furthermore, you could add a discussion section in which you will critically discuss and synthesise all your findings.

Response: Thank you for this comment. In the revised manuscript, we added a section on Methodology and Literature search and Discussion as your suggestion.

We hope that we have addressed all editor’s and reviewers’ concerns and that this manuscript provides additional knowledge regarding vitamin D and visceral obesity in humans. We appreciate your consideration. Please do not hesitate to contact us should you have any further questions.

Sincerely yours,

Hataikarn Nimitphong, M.D.

Reviewer 2 Report

The authors make an interesting and relevant literature review about the effect of vitamin D on adipose tissue and its modulation in obesity with current evidence from preclinical and clinical studies in human adipose tissue.

Major Comments:

Include a section of the search methodology used, although it is understood that this review is not a systematic review or a scoping review, including this information will allow assessing the quality of the selection criteria applied to select the most relevant information.

Include please the Boolean operators of the search for the information consulted (keywords), the temporality of the search (Years), repositories of articles that were reviewed, types of studies selected and included, as well as quality criteria applied to select the information used for the writing of this article. To provide the best scientific evidence of the information shown.

In the section "Role of vitamin D in inflammation" it would be interesting for the authors to describe the extrarenal sources of calcitriol, for example it has been described that the immune cells are capable of expressing CYP27B1 and modulating the synthesis of calcitriol, which may have an immunomodulatory effect on different cytokines. Include this information.

Minor comments:

Beginning in paragraph 86, describe the Vitamin D binding protein with its most described acronym in the literature VDBP instead of DBP.

Paragraph 236, omit the term high sensitivity, because it refers to the sensitivity of the method, but the molecule it refers to is CRP

Author Response

July 18, 2022

Dear reviewer,

We greatly appreciate the opportunity to revise and resubmit our manuscript “Vitamin D and Visceral Obesity in Human: What Should Clinicians Know?”

We have carefully considered useful suggestions from the Editor and Reviewers, and revised the manuscript accordingly. We have provided detailed responses to the Reviewer’s comments below.

Journal: Nutrients (ISSN 2072-6643)

Manuscript ID: nutrients-1811554

Type: Review

Title: Vitamin D and Visceral Obesity in Humans: What Should Clinicians Know?

Authors: Prapimporn Chattranukulchai Shantavasinkul , Hataikarn Nimitphong *

Reviewer 2 comment

Major Comments:

Include a section of the search methodology used, although it is understood that this review is not a systematic review or a scoping review, including this information will allow assessing the quality of the selection criteria applied to select the most relevant information.

Include please the Boolean operators of the search for the information consulted (keywords), the temporality of the search (Years), repositories of articles that were reviewed, types of studies selected and included, as well as quality criteria applied to select the information used for the writing of this article. To provide the best scientific evidence of the information shown.

Response: Thank you for this comment. In the revised manuscript, we added a section on Methodology and Literature search. We included the Boolean operators of search (keywords), the year and the types of studies selected as your suggestion. We also added the limitation of this review in the Discussion.

In the section "Role of vitamin D in inflammation" it would be interesting for the authors to describe the extrarenal sources of calcitriol, for example it has been described that the immune cells are capable of expressing CYP27B1 and modulating the synthesis of calcitriol, which may have an immunomodulatory effect on different cytokines. Include this information.

Response: Thank you for this comment. In the section "Role of vitamin D in inflammation", we added the following sentences “It has been discovered that all cells involved in the innate and adaptive immune system express VDR and CYP27B1, and modulate the synthesis of calcitriol locally, which may have an immunomodulatory effect on different cytokines”, as your suggestion.

Minor comments:

Beginning in paragraph 86, describe the Vitamin D binding protein with its most described acronym in the literature VDBP instead of DBP.

Response: Thank you and we have changed DBP to VDBP.

Paragraph 236, omit the term high sensitivity, because it refers to the sensitivity of the method, but the molecule it refers to is CRP

Response: Thank you and we have changed high sensitivity C-reactive protein (hs-CRP) to CRP.

We hope that we have addressed all editor’s and reviewers’ concerns and that this manuscript provides additional knowledge regarding vitamin D and visceral obesity in humans. We appreciate your consideration. Please do not hesitate to contact us should you have any further questions.

Sincerely yours,

Hataikarn Nimitphong, M.D.

Round 2

Reviewer 1 Report

Dear authors the manuscript has been substantially improved. I would like to suggest a few more changes: In my opinion you should mention clearly in the methodology that this is a narrative review and add the exclusion criteria.

I would expect that the discussion would include a summarised synthetic approach of your findings. One of the limitations is that this is not a systematic review. 

This manuscript is a resubmission of an earlier submission. The following is a list of the peer review reports and author responses from that submission.

Round 1

Reviewer 1 Report

Generally

The authors reviewed the literature to explore the current evidence on the effect of vitamin D supplementation on the reduction of VAT mass and the effect of weight loss modalities on vitamin D status. It is an important topic, however, it is relatively similar to a published review. https:// doi.org/10.3390/ijms23020956. The review is lacking tables and illustrative figures.

Selected comments

·        It is suggested to present the studies on the role of vitamin D in adipocyte differentiation and adipogenesis in a systematic manner in a table form. It will be better in the presentation of species, methodology, VD/its metabolites, and mechanisms.

·        Similarly it is suggested to present studies about the role of vitamin D in energy homeostasis in a systematic manner.

·        Relationship of vitamin D and energy homeostasis in experimental animals was not mentioned e.g.  https://doi.org/10.3389/fphys.2020.00025

·        Long section was written about the relationship between VD and inflammation, however, authors reported early in the review that the inflammatory issues related to VD were more evident and consistent  

·        Effect of medical and surgical weight loss on vitamin D status will be better if some illustrative figures were added to the review

·        Articles 15, 16,22, 68-70, 71,72, 22,73-76 about the effect of bariatric surgery on VD should be resented systematically in a table form.

·        The section about Vitamin D supplementation and weight / visceral fat loss should be revised by adding more studies

Reviewer 2 Report

The intent of the review article by Prapimporn Chattranukulchai Shantavasinkul & Hataikarn Nimitphong is to summarize the role Vitamin D may play on either positively or negatively impacting visceral adipose depots in humans.  While this concept may be initially considered innovative/novel to readers not familiar with this line of research, the current data surrounding this topic is not only scant, but extremely vague & ambiguous.  Even within the authors' review, they consistently state the data on this topic & related subjects is inconsistent & inconclusive, and on other occasions, use terms like unclear, need to be determined, etc.  With this in mind, it is particularly challenging to draft a thorough review on such a specific topic, making it a main concern.  Other issues are outlined below:

*A review without any figures & tables is extremely rare and so incorporating these items may strengthen the review.  There are multiple instances where adding figures/tables to this review may enhance reader interest.

*There are instances where statements are made without providing appropriate references.  For example, a statement which indicates "most" people's main source of vitamin D is skin exposure to sunlight, should be properly cited.

*The keywords chosen are weak.  Keywords should not include terms/phrases that are already included in the title & abstract.

Round 2

Reviewer 1 Report

dear authos 

unfortunately, I did not find any tables or figures included in the uploaded revised copy of the manuscript. please be guided by the template available on the authors' guidelines and include the figures and tables in their appropriate sites in the manuscript. 

best

Reviewer 2 Report

While the reviewers made some modifications to the review article, the topic and related references are still insufficient to form any strong reasoning on whether there is a positive or negative impact upon visceral obesity and vitamin D, which makes the basis of this review article highly questionable.  Also concerning is the addition of the figure and tables, as I noticed that in the text of the manuscript it has been stated these have been incorporated, they do not appear to be included in the revised manuscript with track changes, making it impossible to critique/comment on these items.  Therefore, these are critical issues that a reviewer and potential reading audience cannot overlook, and so need to be addressed.  While the table/figures may be corrected if they add value to the manuscript, it is certainly possible that the overall subject matter of the review article cannot, based on my prior review comments.